# Influence of Wind on Suspended Matter in the Water of the Albufera of Valencia (Spain)

Juan Soria [1,*] , Miguel Jover [2] and José Antonio Domínguez-Gómez [3]

1    Cavanilles Institut of Biodiversity & Evolutionary Ecology, University of Valencia, 46980 Paterna, Spain
2    Research Group of Aquaculture and Biodiversity, Institute of Animal Science and Technology, Universitat Politècnica de València, Camino de Vera 14, 46071 València, Spain; mjover@dca.upv.es
3    GIS and Remote Sensing, Murcia Institute of Agri-Food Research and Development, C/Mayor s/n, La Alberca, 30150 Murcia, Spain; josea.dominguez@carm.es
*    Correspondence: juan.soria@uv.es; Tel.: +34-649-836-836

**Abstract:** Wind significantly influences suspended matter in lakes, especially in shallow lagoons. To know how wind affects the water in Albufera of Valencia, a shallow coastal lagoon, the measured variables of turbidity and transparency have been correlated with the estimates by processing Sentinel-2 satellite images with the Sen2Cor processor. Data from four years of study of winds show that most of them are light to gentle easterly breezes and moderate to fresh westerly breezes. The obtained results show significant correlations between the measured variables and those obtained from the satellite images for total suspended matter and water transparency, as well as with the average daily wind speed. There is no significant correlation between wind and chlorophyll *a*. Moderate to fresh breezes resuspend the fine sediment reaching concentration values from 100 to 300 mg L$^{-1}$ according to satellite data. However, it is necessary to obtain field data for the values of moderate and fresh winds, as for now, there are no experimental data to verify the validity of the satellite estimates.

**Keywords:** transparency; suspended solids; wind effect; shallow lake; Sentinel-2

## 1. Introduction

Turbidity in lakes is one of the most visual elements that we all notice in the landscape. In the human mind, good water is clear and transparent, while the presence of turbidity has always been considered as an indicator of danger and poor quality [1]. The phrase "clear as water" indicates just that obvious property of water is its transparency. Water is used in many contexts as a test of quality, and is highly valued—for example, in certain places, a property has a higher value if the surrounding lakes are transparent [2].

Turbidity is caused by the presence of dispersed particles in suspension, either organic (such as microscopic organisms, fragments of living beings), or inorganic (such as silts and clays) of autochthonous or allochthonous origin from other places [3]. In lakes, turbidity can be of allochthonous origin, due to the arrival of materials from the basin, or of autochthonous origin, due to the growth of organisms that live suspended in the water forming plankton, or due to the suspension of materials by water movements.

Water currents and wind are the main agents in the movement of water. In the case of lakes and reservoirs, wind is the meteorological phenomenon causing the movement, generating waves, currents, circulation cells, and seiches. In the case of shallow lakes, circulation cells are the most important physical phenomenon. In deep lakes, when the wind blows with values above 30 km h$^{-1}$, the depth of the wind-induced circulation cell reaches up to 8 m depth in about 15 min [4], so the epilimnetic layer is perfectly mixed and all homogeneous. In shallow lakes, given the shallow depth, the circulation cell reaches from the surface to the bottom, causing sediment resuspension, distributing the particles throughout the water column, and producing differential sedimentation as a function of density as soon as the wind force ceases.





In shallow lakes, wind is the most important factor in producing the mixing of their waters and rapid interaction with the thin surface layer of sediment [5]. This causes that in regions where winds are steady and dominant from one direction for a period of several hours, these fine sedimentary materials are easily resuspended, and the water column exhibits different turbidity conditions than calm or breezy periods.

Earth observation satellites have been used since their inception for many uses related to the surface of the planet that can be observed by their optical properties. Since the Landsat series has been in operation, the existing bands have made it possible to separate land areas from water areas, including wet areas. The existing algorithms since Landsat-5 came into service were able to separate clear water from turbid water. The Quick Atmospheric Correction (QUAC) method is the most widely used; it is implemented in the ENVI application (Harris Geospatial Solutions, Broomfield CO, USA), which uses the visible and near-infrared bands to obtain the water surface with suspended matter (water mud).

The interest in turbid and clear water has led the Sentinel series of satellites to design specific bands for various uses, combinations of which are useful for water turbidity detection, using the 443 and 490 nm bands [6,7]. The downwelling diffuse attenuation coefficient $(Kd(\lambda))$, is an apparent optical property that characterizes underwater light fields and is determined by inherent optical property [8].

In the past, due to the complexity of measuring this parameter, in order to summarize and specify its use, studies have been focused on two scopes: (1) In inland water was studied the diffuse attenuation coefficient of photosynthetic active radiation KdPAR [9], and (2) in ocean water has been the coefficient at 490 nm (Kd490) [10]. Besides, in inland water, KdPAR is related to the Secchi Disk (SD); being that the product the KdPAR and SD is constant [9].

The multispectral images from satellites have unified the study of the optical properties in inland water, and we can find studies in which the Kd490 is the parameter used. We compared images that the sensor designed to study ocean water, such as MODIS (Moderate Resolution Imaging Spectroradiometer) and OLCI (Ocean and Land Color Instrument) to study inland water [11]—although these are more used the KdPAR or SD and its relationship with the turbidity [12].

Inland water studies that use remote sensing have been a secondary topic in the field of remote sensing, due to the interest from more influential and lucrative fields, such as agriculture. However, sensors designed for agriculture have been exploited by inland water studies, such as the Sentinel-2 program. Sentinel-2A and -2B was designed for agriculture monitoring, and they have a band at 490 nm [7]. The use of this band in freshwater may be used to estimate the Kd490 in relation to other parameters, such as SD, and turbidity with a good spatial resolution, e.g., 10 m.

The measurement of chlorophyll *a* by satellite was one of the first variables considered, since satellites have bands that are sensitive to the radiometry of this pigment, such as the red band, as well as the green band, for uses related to agriculture. Therefore, in the first works, despite the difficulties of the low reflectance in waters, initial studies were carried out on pigments in coastal waters [13], and also in the Albufera of Valencia [14].

From these, automated calculation of clear water pixel identification and calculation of total suspended matter (TSM) and light attenuation depth at 90% (KD_z90max) are implemented in image management applications, such as SNAP (Brockmann Consult, Hamburg, Germany). The results of this application are presented in several works performed both using Landsat-8 and Sentinel-2 on various shallow lakes [15–18], both individually and combining both sensors [19,20]. In other studies, green and red bands have also been used [21]. Water quality parameters (such as TSM, KD_z90max, chlorophyll *a* concentration) are based on the evolution of the Case 2 Regional Coast Color Neural Network (C2RCC) [7]. The C2RCC Neural Network has a long history; its origin is related to the MERIS products (Medium Resolution Imaging Spectroradiometer, an instrument aboard the Environmental Satellite), as well as the Case-2 Regional Processor developed by Freie

Universitat Berlin. The MERIS plug-ins were based on several field data campaigns in Case II waters [22].

The Albufera of Valencia is a shallow coastal lagoon located on the coast of the Mediterranean Sea, between the mouths of the rivers Turia to the north and Júcar to the south, residue of the old wetland that extended in this area, nowadays transformed into orchard crops and rice fields; its origin is in the closure of a marine bay by the growth of a sandy bar of fluvial origin about 3000 years ago [23]. It is located about 10 km S of the city of Valencia (Figure 1). Due to the important presence of migratory and nesting waterfowl, the territory occupied by the Albufera and the peripheral rice field was declared Special Protection Areas for birds in 1976, Natural Park in 1986, Ramsar site in 1989, and was included in the EU's Natura 2000 Network list [24].

The ecological state of the Albufera lagoon by remote sensing has been studied since the first works with Landsat-5 and very recently with Sentinel-2, as well as turbidity [25], demonstrating the correlation of Sentinel-2 measurements and field data, which are closely associated with the water practices of rice cultivation, and the management of the floodgates of communication with the sea. But these works have only used field data taken at times of light and gentle winds. It would be convenient to know the effect of the moderate and fresh wind on the presence of suspended solids in the water, coming from the sediment, in order to consider possible future management measures to facilitate their evacuation to the sea, and therefore, the present work was proposed to assess the correlation between wind speed and total suspended matter (TSM) in the water of the lagoon. The objective of this work is to study how wind affects the increase in TSM concentration in the water of the Albufera of Valencia and its possible relationship with transparency and chlorophyll *a*, based on measurements made with images from the multispectral sensor (MSI) of the Sentinel-2 satellite.

## 2. Materials and Methods

The Albufera of Valencia is a shallow and hypertrophic lagoon. Its characteristics are well described in the works of Soria [24], and other more recent publications [23,26]. It is 23.1 km² (the second largest coastal lagoon in Spain), with an average depth of 0.9 m (maximum of 1.6 m). It is a fresh lagoon, due to the regulation of its connection with the sea, and it has been in a bad ecological state since 1972, according to the quality variables used by the Water Framework Directive and the TSI trophic status index [27]. Its bad trophic state is due to urban and agricultural pressure that contributes nutrients to the inflows.

According to studies of the Albufera sediment described by Marco et al. [28], the lagoon was brackish until the 18th century. He describes that the lithological materials are sandy-mud in the first 10–20 cm and sandy below. The current sedimentation rate is between 4 and 6 mm per year, whereas 200 years ago, it was 1 mm. The decrease in the lagoon surface in the last 200 years has been due to the occupation of land by rice fields.

The meteorological data for the study were collected from the station called *Tancat de la Pipa* located on the very edge of the lagoon (belonging to the Valencian Association of Meteorology, Figure 1); it is an automatic station that records data in real-time and stores statistical values in the network data server. The wind classification (Table 1) has been made according to the Beaufort scale designations [29].

**Table 1.** Equivalence between the Beaufort scale, wind speed, and standard designation in the studied interval.

| Beaufort Scale | Wind Speed (km h$^{-1}$) | Designation |
|:---:|:---:|:---:|
| 1 | 0–4 | Light air |
| 2 | 4–11 | Light breeze |
| 3 | 11–18 | Gentle breeze |
| 4 | 18–25 | Moderate breeze |
| 5 | 25–32 | Fresh breeze |
| 6 | 32–40 | Strong breeze |

Transparency field data were determined from a boat at various sampling points in the lagoon, covering its possible spatial heterogeneity, measuring the depth of vision of the Secchi Disk (SD) of 20 cm diameter. The turbidity of the waters was measured from a vertical sample collected using a hydrographic bottle integrating between 10 and 50 cm in depth, from which an aliquot is collected and transferred refrigerated and in darkness to the laboratory; the determination is performed in a spectrophotometer (Beckman DU600, Beckman Coulter, Brea CA, USA) by absorption at 580 nm, contrasting the values with a curve of standards between 0 and 100 Formacine Absorption Units (FAU) [30]. The determination of TSM was measured as dry-weight and loss of ignition (LOI) of the sample filtered through a 0.45 μm pore glass filter.

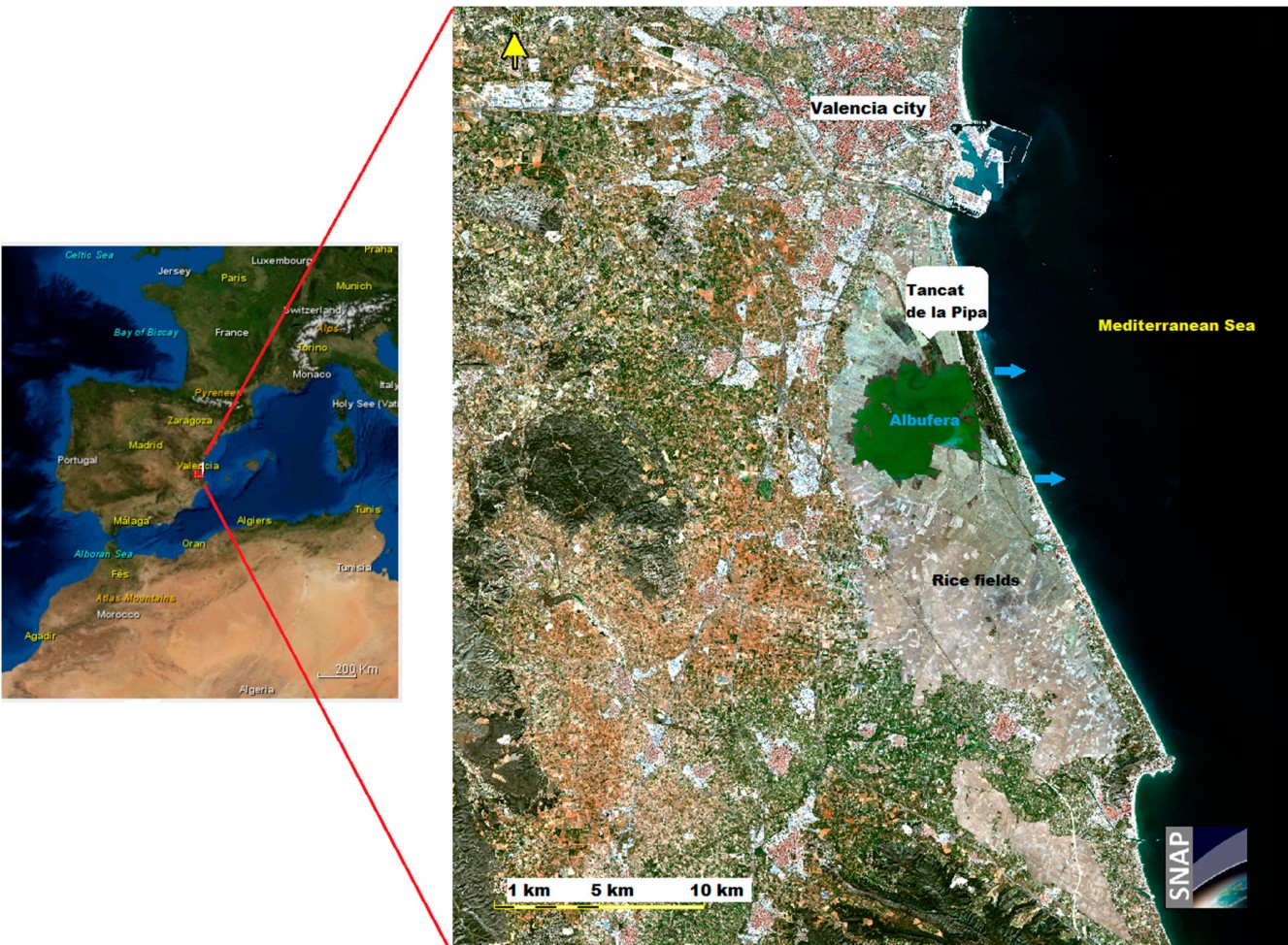

**Figure 1.** Location map of the Albufera of Valencia, indicating with blue arrows the outflow points of the water from the lagoon to the sea. Sentinel-2 false color RGB base image of 12 March 2019.

Remote sensing data have been obtained from studying Sentinel-2 satellite images. The files are downloaded from the ESA Copernicus server (https://scihub.copernicus.eu/dhus/ accessed on: 15 December 2020) in Level 1C format, which is already geographically corrected. The processing is performed using the SNAP application. First, the image is resampled to 10 m pixel size. Then, a subset of the territory under study, the Albufera of Valencia, is cropped using the decimal geographical coordinates 39.383 N, −0.402 W, 39.291 S, −0.285 E. The image obtained is visualized in false color RGB using bands 2 (blue), 3 (green), and 4 (red). The process is then carried out using the Sen2Cor280 tool implemented in SNAP, performing the calculation by the C2X system of neural networks. The processor uses the coastal aerosol, blue, green, and red reflectance bands, in conjunction

with the far-red bands, to perform the atmospheric correction and create a new virtual band in engineering values for the variables total suspended matter (TSM in mg $L^{-1}$), chlorophyll *a* (CHL in mg $m^{-3}$) and the attenuation coefficient of 90% of the diffuse light (KD_z90max, estimated as extinction coefficient in $m^{-1}$). Next, the thematic maps of the results, presented in arbitrary color scale on the background in unprocessed RGB false natural color, are drawn. The results obtained for the variables are processed using the Excel spreadsheet (Microsoft Corporation, Redmond WA, USA) and the statistical software PAST 4.05 [31].

## 3. Results

### 3.1. Meteorological Data

The meteorological data were collected from 1 January 2016 to 31 December 2019. The wind variables were obtained for the daily average in km $h^{-1}$ and the average dominant direction of the day (sectored in the cardinal points). 44 % of the days are windy above 10 km $h^{-1}$. In general, the prevailing winds are westerly, generally in winter and spring; while light breezes are easterly, especially in summer and autumn. Occasionally there are also gentle breezes that can be from the southeast, especially in autumn. The difference lies in the fact that while westerly winds last between three and seven consecutive days, southeasterly winds usually last only a few hours a day, between 13:00 and 23:00 h approximately (except during autumn storms, three or four times a year, which are continuous for one or two days). Figure 2 shows the dominance of easterly and westerly winds, as well as the greater strength of the W winds compared to the E winds.

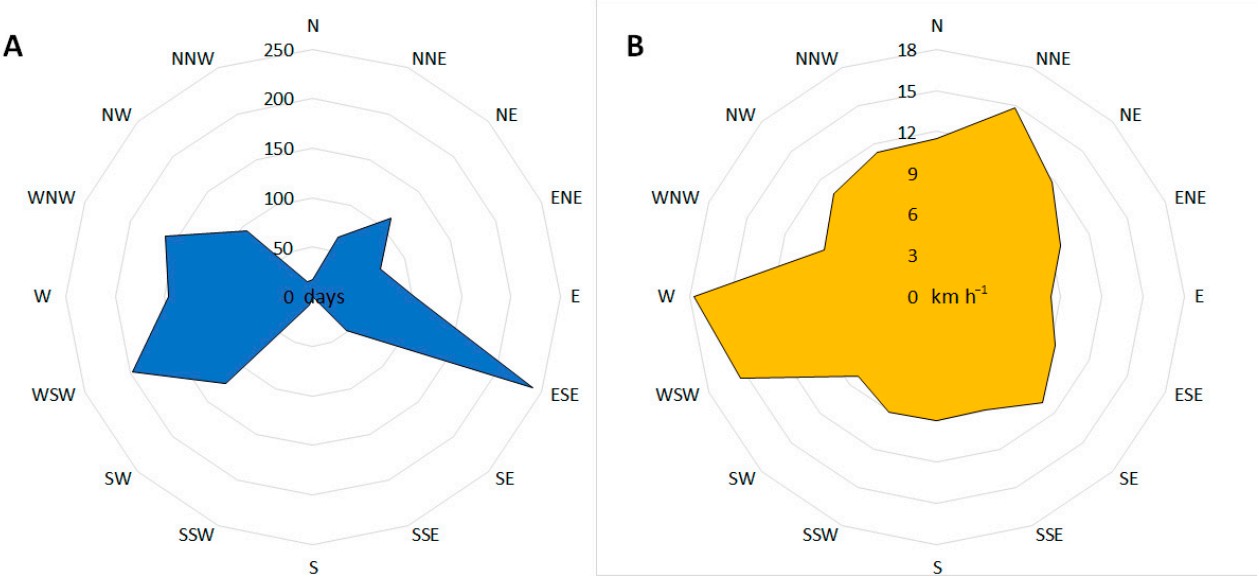

**Figure 2.** (**A**) Days of dominant wind directions during the four study years between 2016 and 2019. (**B**) Average wind speed (km $h^{-1}$) in each of the directions.

Westerly winds are classified as gentle to moderate, with an average value of 17.7 km $h^{-1}$ for those of W origin, while easterly winds are the mildest and would be classified as light breezes, with an average value of 8.3 km $h^{-1}$. Figure 3, on the other hand, shows us how the maximum daily wind values occur mainly in westerly winds and on a few occasions in NE winds, where they would be classified as fresh and strong breezes (between 29 and 38 km $h^{-1}$). The K-means cluster study provides five groups where the first group includes days with fresh to strong breezes (Beaufort force 5 and 6) with speeds above 26 km $h^{-1}$; the remaining four groups account for 95% of the cases, with the second group, including weak to moderate breezes (force 4, speeds between 17 and 26 km $h^{-1}$); The third group includes gentle breezes (force 3 between 11 and 17 km $h^{-1}$), and light breezes are in the fourth

(force 2, between 8 and 11 km h$^{-1}$) and fifth groups (force 2, less than 8 km h$^{-1}$). The daily average winds of force 1 (between 0 and 4 km h$^{-1}$) are very rare, and are considered by the statistical study as atypical, having occurred only on four dates out of the 1461 studied (Figure 3B).

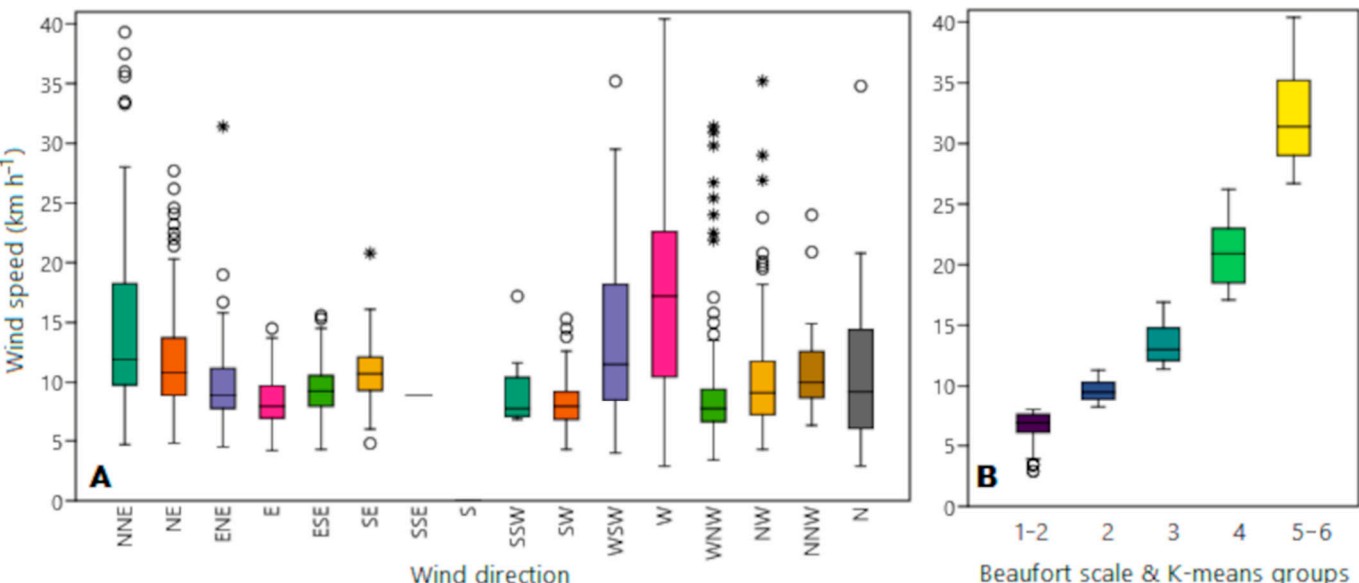

**Figure 3.** (**A**) Average daily wind speed (km h$^{-1}$) in each of the wind rose sectors. The circles and asterisk indicate the outlier values. (**B**) Grouping of values according to K-means and equivalence with the Beaufort scale.

The daily series of wind measurements has been studied, showing that the data are autocorrelated (Figure 4). It is observed that the wind is distributed in cycles, in which there are continuous periods of moderate winds and others of light winds, following a periodicity.

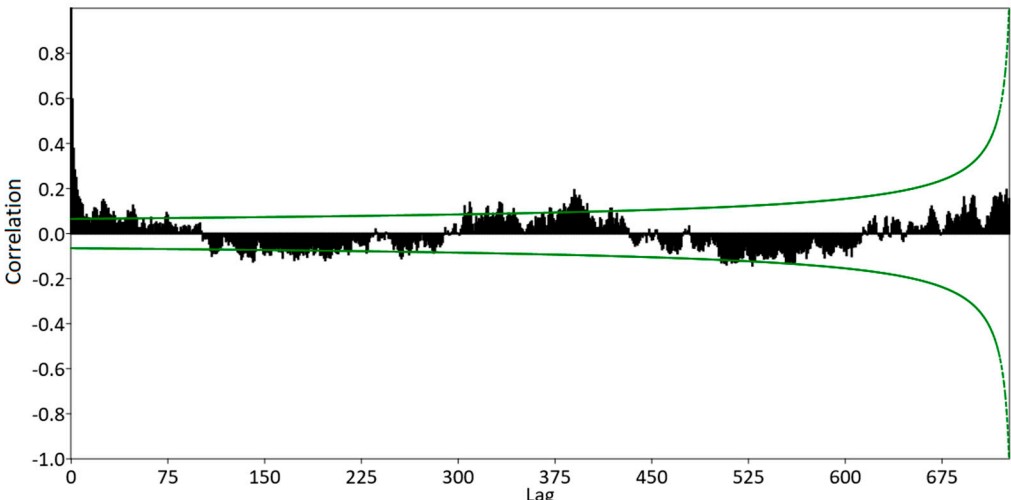

**Figure 4.** Autocorrelation analysis of the wind time series plotted against the mean of the four-year series.

### 3.2. Remote Sensing Images

A total of 27 cloud-free images have been downloaded, allowing processing to obtain valid results for the variables used in this work. The date range has been from January 2016 to April 2019, and the specific days are shown in Table 2. The coincidence of images

has been sought during windy days especially, having obtained 12 images in a period of winds above 10 km h$^{-1}$—of which only one has been with ESE direction and the remaining with westerly winds. In addition, eight images have been obtained in a period of easterly or variable breezes and seven in gentle westerly winds.

**Table 2.** Date of the selected images, indicating the mean wind speed in the previous 72 h; the dominant direction and the values estimated from the satellite image for total suspended matter (TSM), chlorophyll *a* concentration (CHL), and the diffuse attenuation coefficient of light penetration (KD_z90max). High (poor) values are presented in warm colors and low (good) values in cool colors.

| Date | Wind km h$^{-1}$ | Direction | TSM mg L$^{-1}$ | CHL mg m$^{-3}$ | KD_z90max m$^{-1}$ |
|---|---|---|---|---|---|
| 16/01/2017 | 27.5 | WSW | 161.62 | 35.19 | 0.34 |
| 05/02/2017 | 27.5 | WSW | 333.80 | 89.99 | 0.11 |
| 25/02/2017 | 6.9 | ESE | 56.70 | 16.74 | 0.72 |
| 17/12/2017 | 12.1 | W | 50.75 | 30.26 | 0.50 |
| 11/01/2018 | 18.5 | WSW | 217.77 | 78.13 | 0.24 |
| 25/02/2018 | 9.3 | VAR | 63.65 | 84.66 | 0.55 |
| 07/03/2018 | 20.9 | WSW | 143.13 | 43.43 | 0.32 |
| 12/03/2018 | 22.6 | WSW | 315.43 | 35.72 | 0.12 |
| 20/06/2018 | 8.6 | ESE | 63.97 | 56.14 | 0.38 |
| 05/07/2018 | 8.9 | NE | 64.99 | 40.75 | 0.41 |
| 20/07/2018 | 8.5 | VAR | 57.09 | 34.66 | 0.46 |
| 04/08/2018 | 6.8 | ESE | 54.85 | 27.79 | 0.50 |
| 09/08/2018 | 12.4 | ESE | 51.53 | 28.87 | 0.50 |
| 29/08/2018 | 9.6 | ESE | 61.47 | 27.35 | 0.45 |
| 13/09/2018 | 8.4 | WNW | 53.80 | 37.45 | 0.45 |
| 03/10/2018 | 7.1 | WNW | 54.80 | 46.42 | 0.39 |
| 27/11/2018 | 16.9 | WSW | 163.92 | 63.25 | 0.27 |
| 12/12/2018 | 8.1 | WSW | 75.90 | 46.27 | 0.35 |
| 17/12/2018 | 18.7 | WSW | 111.28 | 67.32 | 0.24 |
| 01/01/2019 | 6.0 | SW | 84.78 | 49.05 | 0.25 |
| 06/01/2019 | 7.7 | SW | 76.69 | 39.27 | 0.28 |
| 16/01/2019 | 6.6 | SW | 64.45 | 39.16 | 0.40 |
| 20/02/2019 | 7.5 | E | 62.95 | 45.88 | 0.45 |
| 25/02/2019 | 8.0 | W | 54.89 | 31.50 | 0.55 |
| 17/03/2019 | 10.2 | W | 72.10 | 24.85 | 0.53 |
| 11/04/2019 | 16.8 | W | 65.87 | 46.89 | 0.39 |
| 26/04/2019 | 18.3 | WSW | 96.22 | 71.21 | 0.30 |

The highest daily average wind value in the imaging days has been 27.5 km h$^{-1}$ in westerly component on 5 February 2017, while the lowest daily average corresponded to a light westerly breeze of 6.0 km h$^{-1}$ on 1 January 2019. Table 2 presents the results of the images used and the values corresponding to that date for wind and water variables obtained from studying the images for the area. TSM values range between 51.53 mg L$^{-1}$ and 333.80 mg L$^{-1}$; CHL values are between 16.74 mg m$^{-3}$ and 89.99 mg m$^{-3}$, and light penetration values as KD_z90max are in the range between 0.11 and 0.72 m$^{-1}$.

### 3.3. Multivariate Results

The correlation between mean wind speed and TSM and KD_z90max is statistically significant, presenting coefficients of determination of 0.707 and 0.392, respectively ($n = 27$, $p < 0.001$). The results show that the increase in wind speed increases the total suspended matter, and at the same time, decreases the diffuse attenuation coefficient (Figure 5), i.e., the water becomes more turbid, due to the presence of suspended matter.

The suspended matter was also considered to be influenced by the presence of phytoplankton; however, there is no significant correlation between wind and chlorophyll *a* concentration ($r^2 = 0.178$), nor between suspended matter and chlorophyll *a* ($r^2 = 0.219$). However, suspended matter is significantly correlated with the diffuse attenuation coefficient ($r^2 = 0.579$, $p < 0.001$). In a subset of 59 field samples, turbidity and TSM were measured in the laboratory. The results also show a significant correlation between both variables ($r^2 = 0.42$, $p < 0.001$).

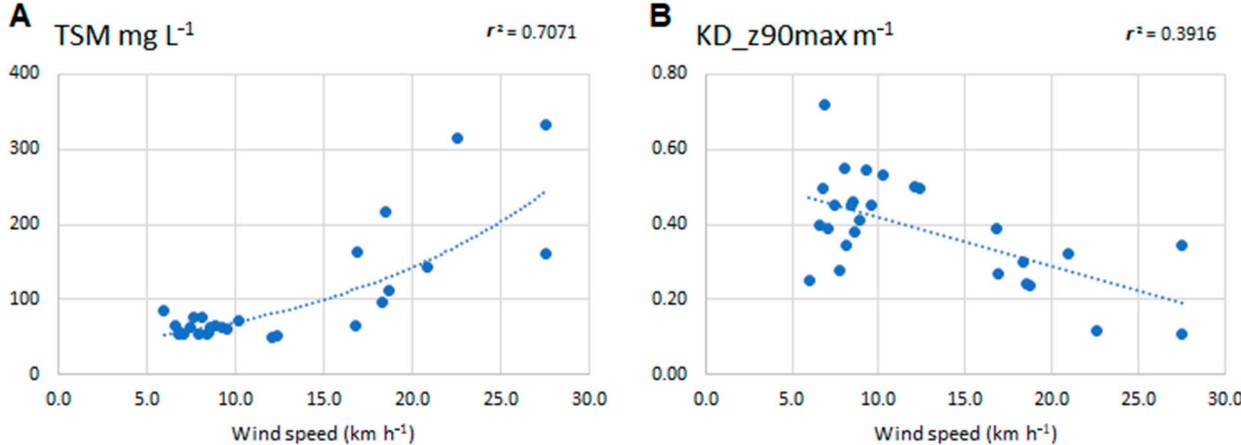

**Figure 5.** (**A**) Correlations between daily mean wind speed and total suspended matter (TSM) and (**B**) diffuse attenuation coefficient (KD_z90max).

The TSM analysis in the laboratory shows that 67% of the dry matter corresponds to organic matter that is lost by ignition (LOI). Ninety-five percent of the samples studied were in the range between 52% and 82% LOI.

Finally, the relationship between the variables estimated by Sentinel-2 images and variables measured in the laboratory was explored. In this case, we considered the pairs between the estimated total solids (TSM) and the turbidity measured in the laboratory and between the diffuse attenuation coefficient and the Secchi disk depth of view. In both cases, we have used 34 individual measurements of water samples collected at an exact point in the lagoon and the value estimated by the satellite image at the same geographic coordinate. Figure 6 presents the linear regression fit lines, which are statistically significant. It should be noted that the relationship between the Secchi disk measurement and the diffuse attenuation coefficient is practically 1:1, as expected since both variables measure the same physical variable.

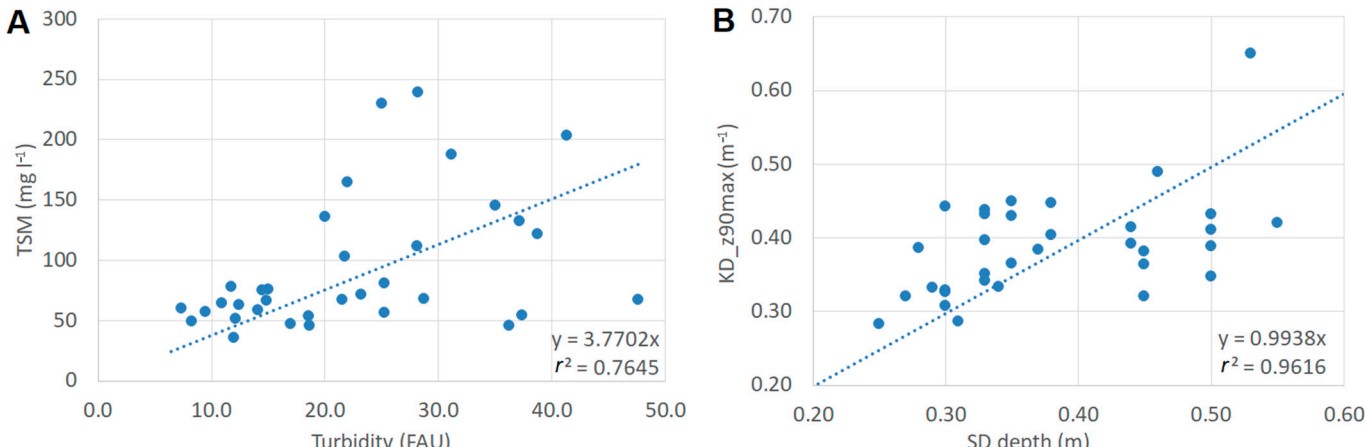

**Figure 6.** For the same geographical point in the lagoon, (**A**) relationships between the values of turbidity measured in the water lagoon and the estimated values of suspended matter (TSM); (**B**) relationships between the Secchi disk (SD) measurement and the diffuse attenuation coefficient (KD_z90max).

The processing of Sentinel-2 images provides us with a spatial view of the distribution of the variables considered at the time of the satellite pass. Figure 7 presents the case of a day of light breeze, and Figure 8 shows a day of moderate breeze.

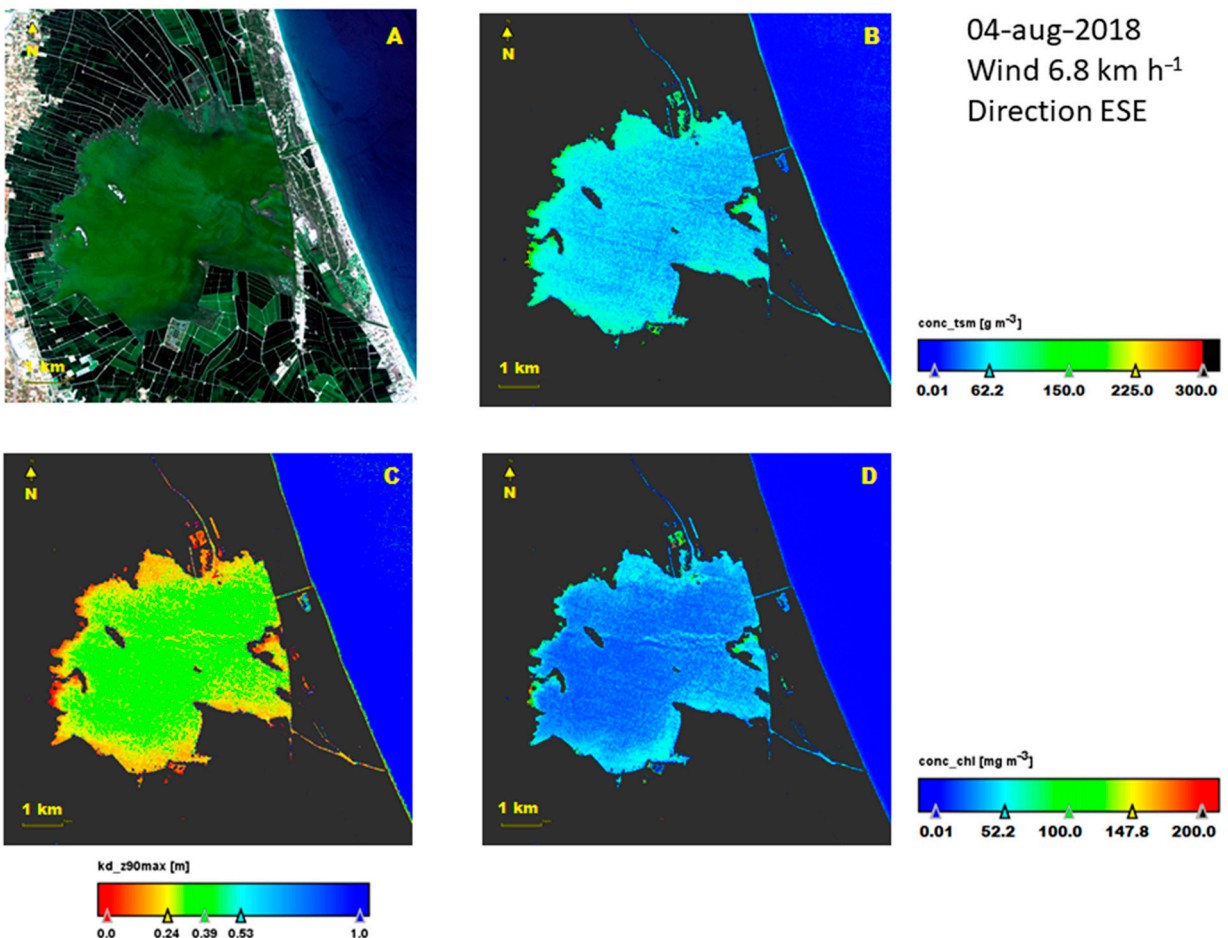

**Figure 7.** Sentinel-2 image and thematic maps of the variables on a day with an easterly light breeze. The rice field is flooded and with overgrown rice plants. (**A**) False color RGB; (**B**) TSM concentration map; (**C**) Transparency map; (**D**) chlorophyll *a* concentration map.

In the image of light breeze (Figure 7), the green tone of the lagoon water in a hypertrophic state, due to the presence of phytoplankton growth can be seen in the false color. The green color is brighter than that of the rice fields surrounding the lagoon. The satellite images show the areas near the shores of the lagoon to be more turbid, with the central open area having the lowest values for all three variables.

In the image with the moderate breeze (Figure 8), the false color of the water shows the grayish tone of the fine materials suspended in the water, as opposed to the greenish color of the areas near the shore, which are more protected from the wind. The rice fields are not cultivated, and the grayish tone of the suspended sediments can also be seen in the water that floods the fields. The variables studied to show the opposite distribution to Figure 7. High turbidity and low transparency values are found in the open water areas of the lagoon, especially to the east and south, while areas of lower turbidity are found near the northern and western margins, sheltered from the wind.

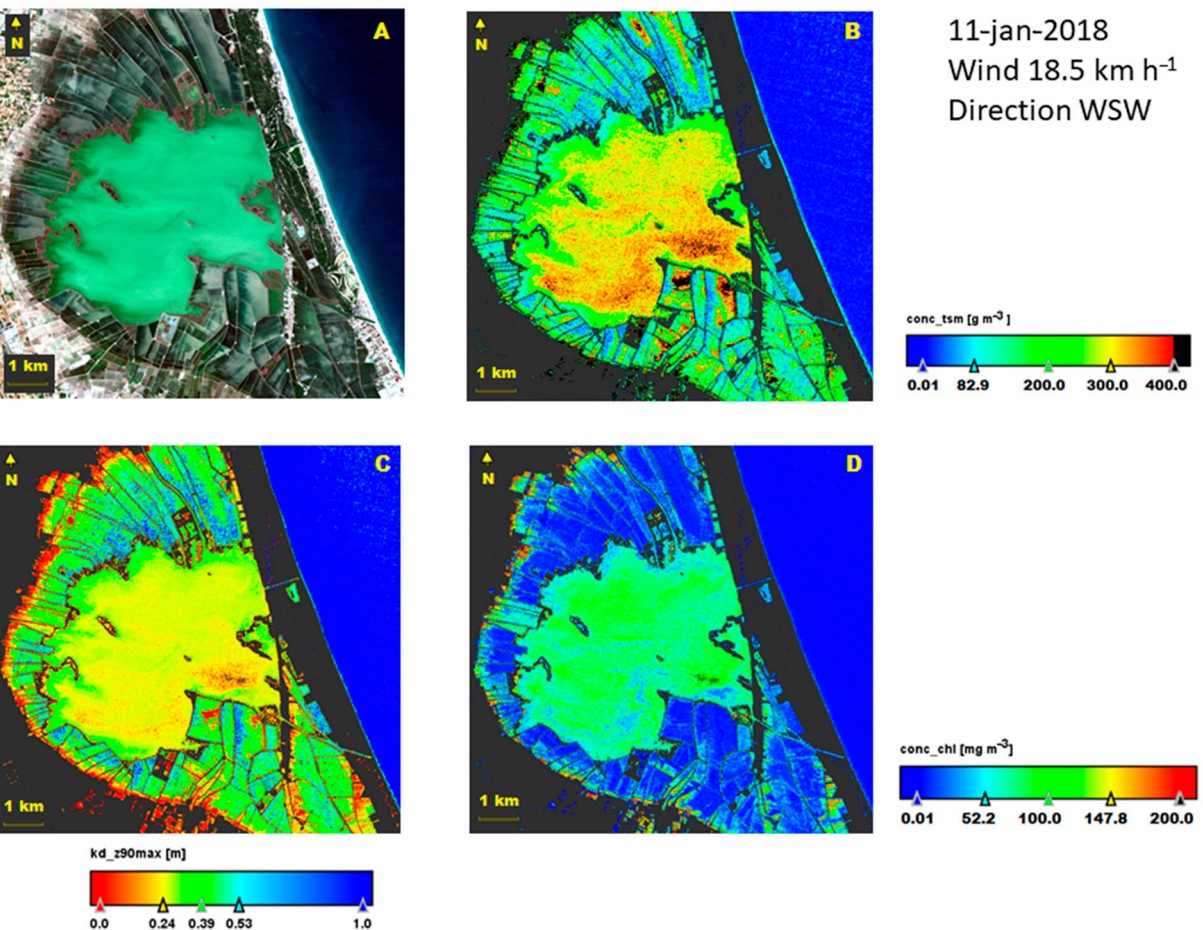

**Figure 8.** Sentinel-2 image and thematic maps on a westerly windy day. The rice field is flooded, but without vegetation. (**A**) False color RGB; (**B**) TSM concentration map; (**C**) transparency map; (**D**) chlorophyll *a* concentration map. Note that the scale of TSM is greater than Figure 7.

## 4. Discussion

The pleasant human perception of an aesthetic judgment in a natural environment includes generalist aspects, such as the quality of the landscape, the complexity of the environment, and also its physical characteristics [32]. The landscape of the Albufera of Valencia is appreciated by the neighbors of the area, as well as by many tourists whose tour includes a visit to the lagoon, especially at sunset (Figure A1). However, the turbid appearance of the water is the factor that most displeases visitors, along with the blooms of cyanobacteria (Figure A2) that add green color to the turbidity [26]. For this reason, efforts to improve water quality and restore transparency have been important for the last thirty years [23,24].

The wind regime of the Albufera is mainly from the east and west, with winds from the north or south being very rare. The location next to the Mediterranean Sea to the east is the decisive factor for the distribution of daytime breezes, generally light, during the central hours of the day. However, the easterly winds become gentle to moderate in the afternoons, turning southeast, known locally as "llebejà". The westerly winds are stronger than the easterly winds and are conditioned by the global meteorology, typical of the passage of Atlantic squalls that usually leave westerly winds for several days (usually three to seven days, then the breezes disappear).

The usual values of total suspended matter in the Albufera are around 50 mg L$^{-1}$, as can be observed at days of light breezes. The variable that contributes to the increase in water turbidity is total suspended matter. The presence of TSM is significantly correlated

with wind strength, due to the shallow depth and the suspension of fine surface sediment material by wind-generated currents, leading to values exceeding 300 mg L$^{-1}$.

One of the uncertainties of the results lies in the fact that sampling cannot normally be carried out safely in the lagoon on days of moderate to the fresh wind and above because of the impossibility of sailing to collect samples. Therefore, it was not possible to calibrate the turbidity values in the high ranges; it is assumed that the calculation performed by the application SNAP is correct and adjusted from the values that were used at the time to perform the measurement equation. As we can see below, the results obtained are similar to other studies in shallow lakes.

Our results are in agreement with the results obtained by So et al. [33] in Apopka Lake (FL, USA), whose surface area is 1.25 km$^2$ and its mean depth is 1.6 m. The initial TSM values in this lake are around 50 mg L$^{-1}$, and its value when winds blow above 10 km h$^{-1}$ increases to reach measurements between 200 and 600 mg L$^{-1}$ when the wind reaches speeds of 36 km h$^{-1}$ and higher. A similar situation occurs in Lake Taihu in China, whose average depth is 1.9 m. According to the results of Jalil et al. [34], the solids concentration is below 50 mg L$^{-1}$ when winds are less than 10 km h$^{-1}$, while when they reach 36 km h$^{-1}$, the value rises up to 260 mg L$^{-1}$. In Iberá Lake (Argentina), whose mean depth is 2.5 m, the results of Cozar et al. [35] also show maximum values of up to 250 mg L$^{-1}$ of TSM when winds are moderate and fresh, assuming that the usual values of TSM in the lake are less than 10 mg L$^{-1}$ because it is a pristine lake, giving importance to the wind as a variable that affects the resuspension of the sediment.

The measurement of water properties related to turbidity and the presence of suspended matter has also been the subject of numerous studies, since the possibility of using remote sensors became available. In Ichkeul Lake (Tunisia), whose mean depth is 1.5 m, both TSM and turbidity have been related to winds in the area [36] using MODIS satellite, and the critical value of wind speed for sediment resuspension has also been found to be 10 km h$^{-1}$, similar to other works already cited [33,35], and concordant with our results, as the increase in TSM is also detected at mean wind values higher than that magnitude.

Observations using the MERIS sensor of the now defunct ENVISAT satellite in Markermeer Lake (Netherlands) [6], whose mean depth is 3.6 m, confirm that the wind at 10 km h$^{-1}$ is responsible for high TSM concentrations in its waters. In this particular case, the base TSM concentration is above 20 mg L$^{-1}$, and when the wind becomes moderate, the concentration increases as well, similar to the other lakes mentioned above. This confirms the independence between the natural suspended matter of a body of water depending on its trophic state, and that which it presents in times of moderate to fresh winds, especially in shallow lakes, such as the Albufera of Valencia. The model studied in Markermeer Lake also confirms our observations that the highest turbidity occurs in the windward areas, while the windward shores are less affected, and therefore, the relatively lower TSM values are observed (Figure 8). In Lake Taihu, with this same sensor, the correlation of mean daily wind with water transparency has also been estimated [37], presenting a coefficient of determination between these variables of 0.80; in the case of the Albufera, the values have been lower (Figure 5), but also significant. It is possible that in the Albufera, the presence of phytoplankton also influences water transparency, given that chlorophyll concentrations are twice as high as in Lake Taihu and water transparency similar.

It is necessary to improve the knowledge of TSM concentrations in the Albufera to assess the possibility of removing suspended matter to the sea when the wind resuspends the sediment. Future work will be oriented towards sampling on days with moderate to fresh winds to correlate the data with the high TSM values, and thus, know the reliability of the measurements obtained from Sentinel-2 with the Sen2Cor processor.

## 5. Conclusions

The prevailing wind in the Albufera of Valencia is mostly from the east or west. The easterly winds are light and gentle breezes, while the westerly winds are usually moderate to fresh breezes, as well as persistent for several days, between three and seven days. The

study shows that there is a significant correlation between the values of the average daily wind speed in the area of the Albufera of Valencia and the concentration of total suspended matter in the water, affecting in such a way that with winds higher than 10 km h$^{-1}$ there is resuspension of the fine sediment material that also decreases the transparency of the water. However, no correlation between chlorophyll *a* concentration and wind has been observed. Since sampling in the lagoon has so far been carried out on days with weak and light winds, there are no data to know the real concentration of suspended matter in the water at those times. Therefore, they have been estimated from the images of the multispectral sensor (MSI) of the Sentinel-2 satellite, whose values calculated with the Sen2Cor processor are valid, given that in other shallow lakes, the concentration values described are similar to those of the Albufera of Valencia. Future work will focus on this validation between experimental data and data estimated by remote sensors.

**Author Contributions:** Conceptualization, M.J.; Data curation, J.S.; Investigation, J.S., M.J. and J.A.D.-G.; Methodology, J.A.D.-G.; Supervision, M.J.; Writing—original draft, J.S.; Writing—review & editing, M.J. All authors have read and agreed to the published version of the manuscript.

**Funding:** This research received no external funding.

**Institutional Review Board Statement:** Not applicable.

**Informed Consent Statement:** Not applicable.

**Data Availability Statement:** Meteorological data are available from public web services (AVAMET). Field data are available from corresponding author.

**Acknowledgments:** Members of the staff of Albufera Natural Park, who contributed to the field works.

**Conflicts of Interest:** The authors declare no conflict of interest.

## Appendix A

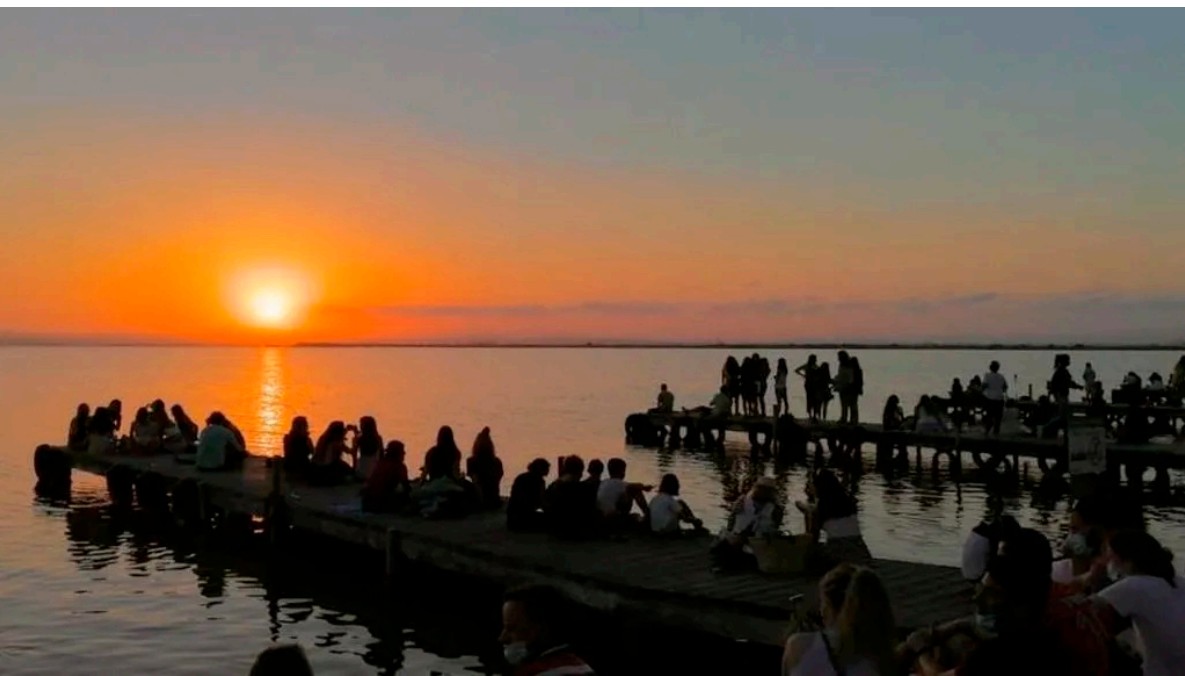

**Figure A1.** Sunset in Albufera of Valencia. Tourists watch the sun lowering on the horizon. Date: 28 August 2020. Source: https://album.mediaset.es/eimg/10000/2020/08/28/clipping_Jv3lJy_aea7.jpg accessed on: 10 March 2021.

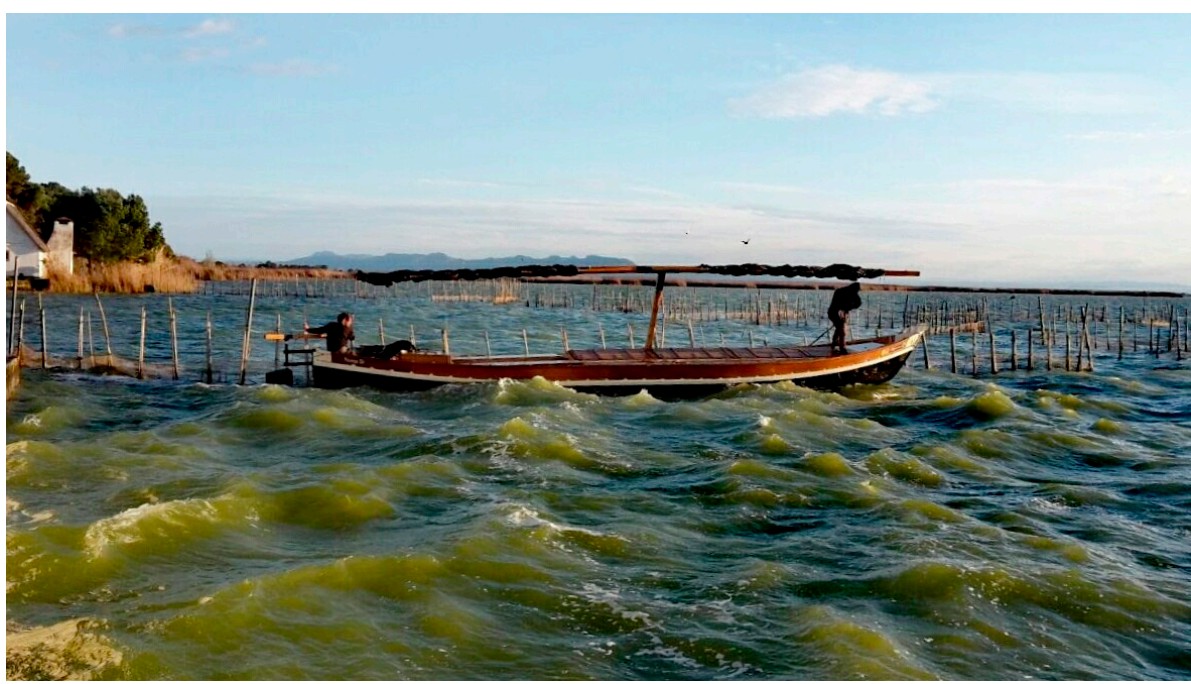

**Figure A2.** Easterly fresh wind about 30 km h$^{-1}$ in Albufera of Valencia. Note greenish water. The 30 cm high waves make the handling of the fishing boat very difficult. Date: 11 April 2018 at 18:30 h. Source: https://www.facebook.com/photo?fbid=847617178773442&set=t.100005755034785 accessed date: 10 March 2021.

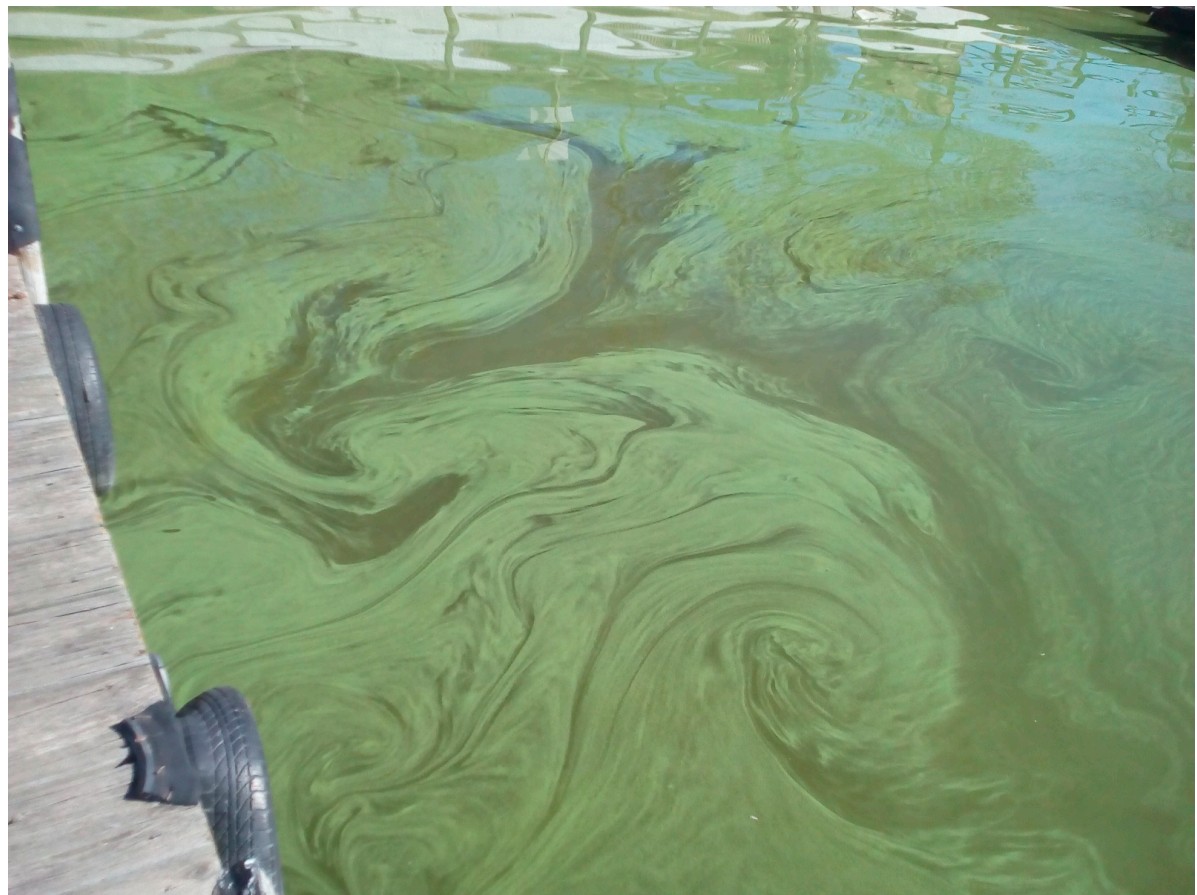

**Figure A3.** Cyanobacteria bloom in Albufera of Valencia. Floating colonies accumulate in some places by breeze effect. Date: 7 November 2016. Source: J.S.

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
