# Peer review of "Influence of Wind on Suspended Matter in the Water of the Albufera of Valencia (Spain)"

_jmse, doi:10.3390/jmse9030343_

Round 1

Reviewer 1 Report

Dear Authors,

Your paper Influence of wind on suspended matter in the water of the Albufera of Valencia (Spain) is interesting and fit the standards for publishing in the JMSE. The highlight of it is the found and cited papers with similar water bodies worldwide, the comparisons made and the conclusions drawn.

My specific comments are:

Line 36,39: change beings with organisms or a similar word.

Line 110: (Although clear) Please insert "North" and "South" since they are used for the first time in the text.

Line 135: Please insert the type of the spectrophotometer, model and manufacturer and move "[26]" at the end of the sentence.

Line 232: Please improve the quality of Figures 2-6, if possible.

Author Response

Your paper Influence of wind on suspended matter in the water of the Albufera of Valencia (Spain) is interesting and fit the standards for publishing in the JMSE. The highlight of it is the found and cited papers with similar water bodies worldwide, the comparisons made and the conclusions drawn.

Thank you very much for your review and comments.

My specific comments are:

Line 36,39: change beings with organisms or a similar word.

Done, we have changed for microorganisms and organisms.

Line 110: (Although clear) Please insert "North" and "South" since they are used for the first time in the text.

Modified.

Line 135: Please insert the type of the spectrophotometer, model and manufacturer and move "[26]" at the end of the sentence.

Incorporated to manuscript.

Line 232: Please improve the quality of Figures 2-6, if possible.

In the final version, the figures are in full resolution. These are draft for the manuscript.

Reviewer 2 Report

I'm not especially familiar with remote sensing.

  • You don't say much about chlorophyll detection in your introduction. Please add some references.
  • In the material and methods section, it is not clear how you get your new virtual bands. Could you explain more? (line 149)
  • You don't explain how you get from FAU to TSM (line 136). An in-lab calibration could be interesting.
  • Concerning the meteorological data in the results section, is wind the only factor that could influence the TSM values? Could there be a previous rain/storm effect? 
  • You make a direct link between turbidity in TSM. This supposes that the SM changes neither in size nor composition. This should be checked on the samples.

A more precise analysis of the TSM characteristics should be made in order to strengthen your results.

Author Response

Thank you very much for your review and comments about the manuscript, that help us to improve the paper.

  • You don't say much about chlorophyll detection in your introduction. Please add some references.

We add a paragraph about the first works in chlorophyll with Landsat-5 (lines 90-94).

  • In the material and methods section, it is not clear how you get your new virtual bands. Could you explain more? (line 149)

This aspect has been improved now: The processor uses the coastal aerosol reflectance bands, blue, green and red, in conjunc-tion with the far-red bands, to perform the atmospheric correction and create a new virtual band in engineering values for the variables.

  • You don't explain how you get from FAU to TSM (line 136). An in-lab calibration could be interesting.

Yes, we have these data measured in laboratory. We add to results: In a subset of 59 field samples, turbidity and TSM were measured in the laboratory. The results also show a significant correlation between both variables (r2 = 0.42, p<0.001).

  • Concerning the meteorological data in the results section, is wind the only factor that could influence the TSM values? Could there be a previous rain/storm effect? 

Yes, if there is a storm or heavy rain, could be an increase of TSM by runoff. But this is very scarce; this is the Mediterranean climate, only 30-40 days of rain in the year. 83 % days of the year it does not rain, but 44 % are windy. We add these in the results.

  • You make a direct link between turbidity in TSM. This supposes that the SM changes neither in size nor composition. This should be checked on the samples. A more precise analysis of the TSM characteristics should be made in order to strengthen your results.

We add this paragraph: The TSM analysis in the laboratory shows that 67% of the dry matter corresponds to organic matter that is lost by ignition. Ninety-five percent of the samples studied were in the range of 52% and 82% LOI.

Reviewer 3 Report

The manuscript submitted for review presents the research on the influence of wind on suspended solids in lake water, especially in shallow lagoons. The objective of this work is to study how wind affects the increase in total suspended matter concentration in the water of the Albufera of Valencia and its possible relationship with transparency and chlorophyll a, based on measurements made with images from the multispectral sensor of the Sentinel-2 satellite.

I believe that the article is written clearly but can be published after minor revision. Below I present a short description of individual chapters with an indication of gaps that should be filled/corrected.

In the introduction, the authors introduce us to the topic of water turbidity in lakes, its sources and the impact of wind on it. The course of satellite research is also presented when it is used to determine the physical parameters of water, especially total suspended matter. The quoted facts are supported by references to literature. Literature review contains items from 1981 to the latest ones from 2020 (items 1-21 of 32). Selective but sufficient selection of literature. The reviewer has no comments.

In Chapter 2, the authors present material and methods. The study area, meteorological data available and remote sensing data from the study of Sentinel-2 satellite images are presented. The cited is literature from 2001-2020 (6 out of 32 from the list of references). The reviewer has no comments.

In Chapter 3, the authors present the result. This chapter describes details of the presented meteorological data, remote sensing images (27 cloud-free images) and multivariate results (the relationship between the variables estimated by Sentinel 2 images and variables measured in the laboratory was explored). The reviewer has no comments.

In Chapter 4, the authors present discussion. The authors presented the results in the field of wind regime and values of total suspended matter. The authors conduct the discussion citing data from the literature. The reviewer has no comments.

At the end of the manuscript in chapter 5, the authors present the conclusions from the work. The reviewer has no comments. Conclusions appropriate.

References: 32 literature items, selected selectively but properly. Is it appropriate to provide literature that is not translated into English? It seems to me that it needs to be improved.

Author Response

Thank you very much for your review and comments about the manuscript, that help us to improve the paper.

About the references in Spanish, there are two works of interest in Spanish, but if the academic editor disagree with it, we can change for another references about the scope in English. We have added also five references more.

Reviewer 4 Report

This paper characterizes the role of winds in resuspension in a small lagoon on the Mediterranean coast of Spain. The study is nice, it employs remote sensing techniques and personal observations, and the number of sources cited is enough. What I appreciate especially is the well-done comparison of the authors' results to the outcomes of the other studies in the other places of the world. The paper is informative and generally well-written. After its careful examination, I see only some minor issues for amendments.

  • Introduction: please, explain the general (international) importance of the Albufeira lagoon.
  • Materials and Methods: please, provide URL(s) to the source satellite images.
  • I suggest to add a few phrases to the lagoon description. First, you need to consider its sedimentary model (the rate of sedimentation, the origin of sediments, etc.). Second, you need to explain what is phytoplankton in this lagoon. Third, what is the anthropogenic factor in this lagoon evolution, as it is located near the urban area (you mentioned connection to the sea, but what about pollution, recreation pressure, etc.?).
  • What are the composition and the origin of the fine sedimentary material that is resuspended by winds in this lagoon? Is it possible to find there some dust from Sahara (often brought to Europe by wind transport)? This would be good addition to Discussion.
  • In Introduction, you state that the turbidity influences the aesthetic properties of the lakes. This is great! Why not to discuss your findings in this direction? In this case, you also need to search and to cite some basic literature sources on landscape aesthetics (e.g., look at this paper: https://www.sciencedirect.com/science/article/abs/pii/S0261517713002185)
  • Please, check the consistency of the terminology use. For instance, TSM is first mentioned in Results, whereas this is important parameter to be considered already in the methodological section. I think all basic terms should appear before Results.
  • The writing is ok, although slight polishing is necessary. I think the authors can do this themselves after reading twice the revised version before it resubmission.

Author Response

This paper characterizes the role of winds in resuspension in a small lagoon on the Mediterranean coast of Spain. The study is nice, it employs remote sensing techniques and personal observations, and the number of sources cited is enough. What I appreciate especially is the well-done comparison of the authors' results to the outcomes of the other studies in the other places of the world. The paper is informative and generally well-written. After its careful examination, I see only some minor issues for amendments.

  • Introduction: please, explain the general (international) importance of the Albufeira lagoon.

A paragraph was moved from Methods to Introduction and added: Due to the important presence of migratory and nesting waterfowl, the territory occupied by the Albufera and the peripheral rice field was declared Special Protection Areas for birds in 1976, Natural Park in 1986, Ramsar site in 1989, and was included in the EU's Natura 2000 Network list.

  • Materials and Methods: please, provide URL(s) to the source satellite images.

Added.

  • I suggest to add a few phrases to the lagoon description. First, you need to consider its sedimentary model (the rate of sedimentation, the origin of sediments, etc.). Second, you need to explain what is phytoplankton in this lagoon. Third, what is the anthropogenic factor in this lagoon evolution, as it is located near the urban area (you mentioned connection to the sea, but what about pollution, recreation pressure, etc.?).

We add this paragraph: According to studies of the Albufera sediment described by Marco [28], the lagoon was brackish until the 18th century. He describes that the lithological materials are sandy-mud in the first 10-20 cm and sandy below. The current sedimentation rate is between 4 and 6 mm per year, whereas 200 years ago it was 1 mm. The decrease in the lagoon surface in the last 200 years has been due to the occupation of land by rice fields.

  • What are the composition and the origin of the fine sedimentary material that is resuspended by winds in this lagoon? Is it possible to find there some dust from Sahara (often brought to Europe by wind transport)? This would be good addition to Discussion.

We stated that suspended matter has no significant correlation with chlorophyll a (r2 = 0.219). (line 255). The fine material is 2/3 organic matter from inputs into the lagoon from canals and from the decomposition of autochthonous living organisms.

We add in results (line 260-262):

The TSM analysis in the laboratory shows that 67% of the dry matter corresponds to organic matter that is lost by ignition (LOI). Ninety-five percent of the samples studied were in the range between 52% and 82% LOI.

  • In Introduction, you state that the turbidity influences the aesthetic properties of the lakes. This is great! Why not to discuss your findings in this direction? In this case, you also need to search and to cite some basic literature sources on landscape aesthetics (e.g., look at this paper: https://www.sciencedirect.com/science/article/abs/pii/S0261517713002185)

Thank you for this paper, very interesting. I have referenced it at the beginning of discussion now.

  • Please, check the consistency of the terminology use. For instance, TSM is first mentioned in Results, whereas this is important parameter to be considered already in the methodological section. I think all basic terms should appear before Results.

This is a good point. We introduce the term now in the Introduction (see new lines 96, 101, 116, 117) and Methods (lines 151, 169).

  • The writing is ok, although slight polishing is necessary. I think the authors can do this themselves after reading twice the revised version before it resubmission.

Thank you very much, we have revised the manuscript and corrected some mistakes.